# Comparison of Early Contrast Enhancement Models in Ultrafast Dynamic Contrast-Enhanced Magnetic Resonance Imaging of Prostate Cancer

**DOI:** 10.3390/diagnostics14090870

**Published:** 2024-04-23

**Authors:** Alfredo Clemente, Guerino Selva, Michael Berks, Federica Morrone, Aniello Alessandro Morrone, Michele De Cristofaro Aulisa, Ekaterina Bliakharskaia, Andrea De Nicola, Armando Tartaro, Paul E. Summers

**Affiliations:** 1Radiology Unit, Centro Medicina Nucleare N1, “Centro Morrone”, 81100 Caserta, Italy; alf.clemente@hotmail.it (A.C.); rinoselvamd@gmail.com (G.S.); 2Quantitative Biomedical Imaging Laboratory, Division of Cancer Sciences, University of Manchester, Manchester M13 9PL, UK; michael.berks@manchester.ac.uk; 3Radiology Unit, Centro Radiologico Vega, “Centro Morrone”, 81100 Caserta, Italy; federica.morrone92@gmail.com (F.M.); aniellomorrone93@gmail.com (A.A.M.); 4Department of Biotechnological and Applied Clinical Sciences, University of L’Aquila, 67100 L’Aquila, Italy; michele.dec.au@gmail.com; 5Department of Neuroscience and Imaging, University G. d’Annunzio, 66100 Chieti, Italy; ekaterina.bliakharskaia@unich.it; 6Radiology Unit, SS. Annunziata Hospital, ASL Lanciano Vasto Chieti, 66100 Chieti, Italy; andrea.denicola@unich.it; 7Department of Clinical, Oral Sciences and Biotechnology, University “G. d’Annunzio”, 66100 Chieti, Italy; armando.tartaro@unich.it; 8MRI Unit, Santissima Trinità Hospital, ASL Pescara, 65026 Popoli, Italy; 9QMRI Tech, 65123 Pescara, Italy

**Keywords:** prostate cancer, ultrafast dynamic contrast enhanced MRI, pharmacokinetic model, empirical mathematical model

## Abstract

Tofts models have failed to produce reliable quantitative markers for prostate cancer. We examined the differences between prostate zones and lesion PI-RADS categories and grade group (GG) using regions of interest drawn in tumor and normal-appearing tissue for a two-compartment uptake (2CU) model (including plasma volume (v_p_), plasma flow (F_p_), permeability surface area product (PS), plasma mean transit time (MTT_p_), capillary transit time (T_c_), extraction fraction (E), and transfer constant (K_trans_)) and exponential (amplitude (A), arrival time (t_0_), and enhancement rate (α)), sigmoidal (amplitude (A_0_), center time relative to arrival time (A_1_ − T_0_), and slope (A_2_)), and empirical mathematical models, and time to peak (TTP) parameters fitted to high temporal resolution (1.695 s) DCE-MRI data. In 25 patients with 35 PI-RADS category 3 or higher tumors, we found F_p_ and α differed between peripheral and transition zones. Parameters F_p_, MTT_p_, T_c_, E, α, A_1_ − T_0_, and A_2_ and TTP all showed associations with PI-RADS categories and with GG in the PZ when normal-appearing regions were included in the non-cancer GG. PS and K_trans_ were not associated with any PI-RADS category or GG. This pilot study suggests early enhancement parameters derived from ultrafast DCE-MRI may become markers of prostate cancer.

## 1. Introduction

Dynamic contrast-enhanced magnetic resonance imaging (DCE-MRI) is a recommended acquisition in the current Prostate Imaging Reporting and Diagnosis System (PI-RADS v2.1) multiparametric MRI (mpMRI) protocol guidelines for prostate cancer (PCa) imaging [1]. DCE-MRI, however, makes only a secondary contribution to PCa diagnosis under PI-RADS, influencing outcome only when there is focal peripheral zone (PZ) enhancement earlier than or contemporaneous to the surrounding tissue co-located with a DWI and/or T2-suspicious lesion. This suggests that there is room for DCE-MRI to have a more substantial role if it can provide objective indices of prostate disease.

The PI-RADS guidelines adopt this approach because pharmacokinetic (PK) modeling and curve shape analysis of prostate DCE-MRI have been unable to demonstrate robust improvements in diagnostic performance in studies replicated across centers’ imaging [1]. Two possibilities that may be responsible for this lack of reproducibility are, first, that the PK models being used are not appropriate for the physical conditions encountered in prostate cancer and, second, that the temporal resolution of the DCE-MRI time series is not sufficient to reliably differentiate pathological alterations. The majority of reports in the PCa literature have made use of Tofts and extended Tofts models (T and ET model, respectively) with a temporal resolution of more than 5 s per frame (see summaries in [2,3]), entirely consistent with the PI-RADS recommendations suggesting a minimum temporal resolution of 15 s per frame [1]. The T and ET models are widely applied to clinical DCE-MRI, including PCa, but are both appropriate only where the tissue is weakly vascularized (plasma volume (v_p_) near zero) or, in the case of the ET model, highly perfused (large plasma flow (F_p_)) [4]. Alternative pharmacokinetic models, such as the two-compartment exchange model and the two-compartment uptake (2CU) model, have been proposed to overcome these shortcomings. As with the T and ET models, these models consider both the extravascular extracellular and plasma compartment volumes (v_e_ and v_p_, respectively) and the two flow-related parameters (the permeability surface area product (PS) and F_p_) [4]. Similarly, they assume that the tracer distributes over no more than two tissue compartments, whose volumes when normalized by the voxel volume are subject to the constraint v_p_ + v_e_ ≤ 1, that no tracer is lost from the extravascular extracellular space to the environment, and that there is a single inlet to the capillary bed through which the arrival of a tracer can be described by an arterial input function (AIF). These assumptions are plausible in the majority of tissue types, but the presence of necrotic regions or intracellular leakage may violate the assumption of two compartments being present.

One area where the models distinguish themselves is whether or not they consider the possibility that there is substantive bidirectional exchange between the two compartments (Table 1) [5,6,7]. This is generally not an essential concern in the early phase of the DCE-MRI time course, when the back flux of tracer into the plasma space is negligible. In this situation, extraction is effectively unidirectional, and enhancement can be described by the 2CU model that involves only three parameters (v_p_, PS, and F_p_). The 2CU model may also be appropriate for early enhancement in tissues where there are poorly mixed compartments or extraction/convection/diffusion into additional compartments [8]. For a reliable estimation of all three parameters, a good contrast-to-noise ratio (CNR) and high temporal resolution are required [9]. The 2CU model generalizes the well-known Patlak model that is often applied to data with poor temporal resolution. Patlak is applicable whenever the acquisition time is shorter than the contrast agent’s extravascular transit times (T_e_ = v_e_/PS), typically in the range of 2–3 min [10].

Violations of the assumptions underlying a PK model can lead to errors in the fitting of the DCE-MRI data and result in inconsistent or unrealistic parameter estimates. For instance, values of v_e_ > 1 (i.e., greater than the voxel volume) may occur if the native longitudinal relaxation time is incorrect, if v_p_ has not been incorporated (as is the case for the T model) when the plasma volume is not negligible, or if peak enhancement is not reached within the time course [11]. Moreover, most PK model parameters (e.g., v_p_, v_e_, K_trans_, F_p_, and PS) scale with AIF. As a result, errors in AIF arise in the measurement or in changes to AIF due to the dispersion of the bolus between the site of measurement (usually a large feeding artery) and the tissue propagation into parameter errors. The problem of AIF dispersion has been widely discussed in the literature, but so far no generally accepted solution has been found [12].

In light of these difficulties, some authors have foregone attempts to derive bio-physical models of signal enhancement time courses in DCE-MRI [2,13,14], and instead adopt purely mathematical forms that allow greater flexibility and fitting quality. These empirical mathematical models (EMMs) for DCE-MRI avoid the extraction of AIF, directly estimate model parameters without making rigid assumptions about tissue architecture or linear behavior, and can adapt to a wide range of contrast agent kinetics. The parameters of these models, while not necessarily describing specific known biophysical entities, nonetheless characterize the tissue in a given voxel. Unsurprisingly, the stability of EMM fitting benefits from higher temporal resolutions, particularly for models that focus on the initial rate of enhancement [2].

In this work, we evaluate whether parameter values obtained through fitting of the 2CU model, an exponential EMM, a sigmoidal EMM, and a simple curve shape descriptor to the first two minutes of high temporal resolution prostate DCE-MRI differ between regions of tumor and healthy tissue.

## 2. Materials and Methods

This retrospective study involved adult male patients undergoing MRI for suspected prostate cancer between November 2021 and December 2022 for whom subsequent biopsy data were available. The local scientific review board approved the study and waived the requirement for specific consent for data use. The study involved using high temporal resolution DCE-MRI for the determination of enhancement-related parameters for the 2CU model pharmacokinetic model, exponential and sigmoidal EMMs, and simple curve shape metrics from specific tissue regions for comparison with prostate with reference standard biopsy histopathology findings.

### 2.1. Patients

Patients provided informed consent for the performance of the MR and biopsy examinations as part of their routine care and specific consent for data publication.

The inclusion criteria were: Adult males between 40 and 85 years of age;With one or more lesions having a PI-RADS category of 3, 4, or 5;Prostate biopsy targeted to the reported lesion(s) within 6 months of the mpMRI examination.

Exclusion criteria:Prior local or systemic treatment for prostate cancer;Examinations performed without injection of contrast agent;Contraindications to MRI.

### 2.2. DCE-MRI Acquisition

According to the PI-RADS v2.1 guidelines, which call for a temporal resolution better than 15 s per dynamic and a 3 mm slice thickness for the DCE-MRI acquisition [1], the DCE-MRI was acquired with an axial 3D gradient echo sequence (differential subsampling with Cartesian ordering (DISCO)) [15] optimized in-house for ultrafast prostate coverage with a 1.695 s temporal resolution and repeated 150 times during and following endovenous administration of gadolinium-based contrast agent (Dotarem, Guerbet SpA, Milan, Italy); 0.1 mmol/kg at 3 mL/s) via a pump (Medtron AG, Saarbrücken, Germany). All DCE-MRI scans were performed in Centro Medicina Nucleare N1, Centro Morrone (Caserta, Italy) on a 3T MR scanner (Signa Pioneer, GE Healthcare Italy Srl, Milan, Italy) with acquisition parameters as listed in Table 2. 

### 2.3. DCE-MRI Processing

The MRI images were exported in DICOM format and converted to nifti format for the analysis. Images in the DCE-MRI time series were motion-corrected using a deformable registration between frames (4D Elastix extension in 3D Slicer v4.11).

A manually defined ROI was positioned in the left iliac artery in order to obtain a patient-specific temporal representation for arterial input function (AIF(t)). This AIF(t) was assumed to represent the plasma concentration arriving in the prostate without amplitude correction or scaling for the calculation of 2CU model parameters. 

The mean signal (S_0_) was calculated for each voxel over a baseline interval that excluded the first two timepoints and ended with the final frame prior to arrival of contrast in the iliac arteries. The DCE-MRI signal for each voxel was then converted to a contrast agent concentration time course (C(t)) using the change in signal intensity relative to the baseline signal as described by:(1)$$C(t)=R{[S\left(t\right)-S}_{0}]/S_{0}$$
where the constant R depends on the contrast agent relaxivity (r_1_) and the native longitudinal relaxation time (T_10_) (R = 1/(r_1_T_10_)) [16]. Since T1 mapping was not performed in this study, we set R = 1, assuming the approximation of an identical T_10_ for both cancer lesions and healthy prostatic tissues.

An existing bi-exponential model in the MADYM software package (v4.22.1, QBI Lab, University of Manchester, https://github.com/michaelberks/madym_cxx, accessed on 25 May 2023) [17] was adapted to form a 2CU model (Formula (2)) [8] (ch2.5). The biexponential model parameters (α_+_, α_−_, β_−_) were obtained by fitting and used to calculate the 2CU model parameters v_p_, PS, and F_p_ as described in [8] (ch2.3). In turn, these values were used to calculate plasma mean transit time (MTT_p_ = v_p_/(PS+F_p_), capillary transit time (T_c_ = v_p_/F_p_), and extraction fraction (E) [7].
(2)$$C\left(t\right)={{{[\alpha }_{+}+\alpha }_{-}e}^{-\beta_{-}(t-t_{0})}]*AIF(t),$$

Fitting of the 2CU model was limited to the first 2 min from the onset of enhancement and involved minimizing the sum of square residuals between the measured and modeled concentration time courses. Ranges for the fitted parameters were pre-specified (see Table 3) and $\alpha_{+}$ = 0.2, $\alpha_{-}$ = 0.2, $\beta_{-}$ = 4 were used as starting estimates, corresponding to physiological values F_p_ = 0.4 min^−1^, PS = 0.4 min^−1^, and v_p_ = 0.2.

Two EMMs were also fitted to the contrast agent concentration time course C(t) on a voxel-by-voxel basis (BioMap v3.8.0.4, maldi-msi.org), the first having the form of an exponential curve [2,13]:(3)$$C\left(t\right)=A\left(1-e^{-\alpha \left(t-t_{0}\right)}\right),$$
with amplitude (A), enhancement rate (α), and arrival time (t_0_) being obtained. 

The second EMM was a sigmoid curve defined as the integral of the Gaussian probability function:(4)$$C\left(t\right)={\frac{1}{\sqrt{2\pi }}A_{0}\int_{-\infty }^{t}e}^{-\frac{{\left(x-A_{1}\right)}^{2}}{2A_{2}^{2}}}\mathrm{dx},$$where the 3 free parameters being fitted correspond to the amplitude (A_0_), the center time (A_1_), which was expressed relative to the contrast agent arrival time (T_0_) as A_1_ − T_0_, and the slope (A_2_) of the sigmoid for each voxel.

The EMMs were fitted using a weighted non-linear least squares method. The weights of the EMM were 1 for at least the first 120 s of scanning (or to peak of enhancement), after which they were 0 to limit the impact of wash-out on the fitting of the uptake phase described by the EMMs. For the exponential EMM, this cut-off was determined on a voxel-by-voxel basis by the optimization algorithm. Ranges for the fitted parameters were pre-specified (see Table 3) and starting estimates were automatically calculated based on an initial fit to the baseline and ending signal in the time course.

Reflecting the clinical practice, the time to peak (TTP) concentration in the voxel was calculated as a curve shape parameter.

The various parameters obtained with the above procedure and their units of measure are summarized in Table 3, along with ranges of physiologically credible values used to further mask values in the ROIs (described below). We note that for v_p_ we did not impose the plausible physiological upper limit ≤ 1 to take into account scaling issues related to AIF.

### 2.4. Region of Interest Definition

The mean DCE and b = 0 DWI images were co-registered to the T2-weighted images using a mutual information cost function (Jim 8, Xinapse Systems Ltd., West Bergholt, UK), and the resulting affine transformations were applied to the high b-value DWI images, ADC maps, and the DCE timeseries. Working from the radiological report of the mpMRI examination (see below), areas of suspected cancer were identified by visual inspection of these co-registered images. A radiologist manually defined regions of interest (ROIs) on all slices (minimum 2) of the T2 images where lesions having PI-RADS categories of 3 to 5 were visible (3D Slicer v4.11, https://slicer.readthedocs.io). On the same slices, ROIs were also drawn for healthy-appearing peripheral or transition/central zone (PZ or TZ/CZ) or their combination adjacent to the lesion (Figure 1). The ROIs were saved as binary masks in nifti format. The ROIs were affine back-transformed to the DCE-MRI space, and ad hoc scripts were used for the extraction of mean and standard deviation of values from each parameter map (fsltools, fsl ver. 5.6, Analysis Group, FMRIB, Oxford, UK).

### 2.5. Radiological and Pathological Evaluation

All multiparametric MRI examinations were reported according to the PI-RADS v2.1 guidelines by 2 radiologists in the course of routine clinical practice. The lateralization and PI-RADS category of each reported lesion were redacted from the radiological reports. 

Prostate biopsies were performed following the mpMRI examination by a pool of urologists with no less than 5 years of experience using ultrasound/MR fusion guidance. The biopsy procedures were a mixture of template or template + targeted biopsies. The template biopsies involved 3 or more samples per side, while targeted biopsies sought to obtain 2 samples of the target lesion. The pathological analyses of the biopsy samples were performed by a pool of pathologists according to ISUP pathological criteria [18]. The highest grade group (GG) amongst the samples for each side (or the target when taken) were redacted from the pathology reports.

### 2.6. Statistical Analysis

Summary statistics for the patient demographics, radiological description of the lesions, and biopsy pathology findings were calculated in terms of mean and range or count and percentage. 

Four principal analyses were performed for each parameter. The first was a *t*-test comparing the radiologically normal-appearing prostate zones (PZ and TZ). In light of the differences seen between prostate zones for some parameters in this analysis, the second analysis was a two-way ANOVA involving all the radiologically defined ROIs, with PI-RADS category (4 levels: normal-appearing, PI-RADS categories 3, 4, and 5) and the prostate zone (PZ and TZ) as factors. The third analysis was a *t*-test involving only the biopsy samples, reducing the GG results to negative or positive (GG values ≥ 1). To provide a greater number of negative GG results, the fourth analysis was a one-way ANOVA.

ANOVA incorporated the parameter values from the radiologically normal appearing prostate into the GG negative category and separated the positive GG results according to the GG (i.e., 5 levels: GG negative or radiologically normal-appearing, GG1, GG2, GG3, and GG4). In light of the small number of biopsy samples from the transition zone, the third and fourth analyses were limited to the peripheral zone samples. 

Those parameters for which significant differences were observed in the ANOVA analyses were subjected to Tukey’s honestly significant difference (Tukey’s HSD) post hoc test for pairwise comparisons. 

The statistical analysis made use of R (R Core Team (2018), R (language and environment for statistical computing), R (foundation for statistical computing), Vienna, Austria, https://www.R-project.org accessed on 1 October 2023). To ensure a false detection rate below 0.05, a Benjamini–Hochberg correction was performed across the 33 ANOVAs. 

## 3. Results

Of 337 men who underwent a prostate mpMRI examination over the study period, 144 did not agree to participate in this study, 141 had lesions of PI-RADS category 1 or 2, 5 had a DCE-MRI that was severely affected by movement or metal prosthesis, and 22 did not have a prostate biopsy within 6 months or their biopsy result was not available, leaving 25 patients for analysis (Figure 2).

The participants had a mean age of 67 years (range 55 to 84 years) (Table 4). Amongst the 25 patients, a total of 35 clinically significant lesions were identified by the radiologists, of which 11.4% (*n* = 4) were PI-RADS category 3, 60% (*n* = 21) PI-RADS category 4, and 28.6% (*n* = 10) PI-RADS category 5. A majority of the lesions (82.9%, *n* = 29) were in the peripheral zone. The distribution of GG scores was relatively even, with 31.4% (*n* = 11) of biopsies yielding negative findings, and between 11.4% (*n* = 4) and 25.7% (*n* = 9) having GGs of 1 to 4.

Examples of the fitting of the two different EMMs to single-voxel contrast enhancement time courses in a peripheral zone PI-RADS 4 lesion and healthy tissue in a 55-year-old PCa patient are illustrated in Figure 3, and of the 2CU model in a peripheral zone PI-RADS 4 lesion and adjacent healthy-appearing peripheral zone ROI (the same ROIs depicted in Figure 1) in Figure 4.

Example maps of the 2CU model parameters F_p_ and MTT_p_, exponential EMM parameter α, sigmoid EMM parameters A_1_ − T_0_ and A_2_, and the curve shape parameter TTP are presented in Figure 5. The maps of the exponential EMM parameters A and t_0_ and the sigmoid EMM parameter A_0_ showed negligible contrast and did not yield significant differences between regions, and therefore were not included in further analysis. In one subject, the prostate lesion ROI contained no voxels within the target value range for the measured parameter PS (and consequently for the calculated parameters E and K_trans_). This subject was left out of the subsequent ANOVA analyses for these parameters.

Model parameters obtained for radiologically normal-appearing prostate zones are summarized and compared in Table 5. Between the TZ and PZ, only the 2CU model parameter F_p_ (*p* = 0.0245) and the exponential EMM parameter α (*p* = 2.10 × 10^−3^) showed differences, while the 2CU model parameter E showed a trend to significance (*p* = 0.0512).

In the two-way ANOVA (Table 6) considering prostate zone and PI-RADS category as factors (radiologically normal-appearing prostate being assigned a score of PI-RADS 1), F_p_ was no longer seen to differentiate between zones (*p* = 0.403), but α and E, along with the sigmoidal EMM parameters A_1_ − T_0_ and A_2,_ did (*p* = 0.0024, 0.0236, 0.0264, and 0.0321, respectively). Regarding the PI-RADS category on the other hand, significant differences were seen for the 2CU model parameters F_p_, MTT_p_, T_c_, and E (*p* = 2.72 × 10^−5^, 1.45 × 10^−4^, 1.53 × 10^−4^, and 6.99 × 10^−3^, respectively), and for exponential EMM parameter α (*p* = 2.44 × 10^−7^), sigmoid EMM parameters A_1_ − T_0_ and A_2_ (*p* = 4.28 × 10^−6^ and 4.94 × 10^−6^, respectively), and the curve shape parameter TTP (*p* = 4.88 × 10^−7^). In post hoc testing for those parameters showing differences in the above ANOVA, there was a tendency for the model parameters to progress monotonically (either increasing or decreasing) with increasing PI-RADS category (Figure 6 and Appendix A). Thus, differences relative to normal-appearing tissue tended to be larger for higher PI-RADS categories 4 and 5 than for PI-RADS category 3 lesions. This tendency, however, was less pronounced for the transitional/central zone lesions than peripheral zone lesions, likely because of the small numbers of lesions involved.

In the PZ lesions (N = 29), none of the model parameters showed significant difference between those with positive and negative pathology (i.e., GG ≥ 1 vs. reported negative for PCa) (Table 7), and only F_p_ and TTP showed a trend to significance (*p* = 0.0527 and *p* = 0.0607, respectively). Incorporating the radiologically normal-appearing ROI values into the pathology negative samples (to provide a larger sample set for a one-way ANOVA with GG (normal-appearing or biopsy negative vs. GG1, GG2, GG3, and GG4) of the peripheral zone), the same parameters showed associations for GG as for the PI-RADS category (Table 7), namely 2CU model parameters F_p_, MTT_p_, T_c_, and E (*p* = 9.41 × 10^−3^, 0.002, 1.73 × 10^−3^, and 0.0346, respectively), exponential EMM parameter α (*p* = 4.1 × 10^−3^), sigmoid EMM parameters A_1_−T_0_ and A_2_ (*p* = 8.73 × 10^−3^ and 7.03 × 10^−3^, respectively), and the curve shape parameter TTP (*p* = 4.72 × 10^−4^), though the *p*-values were higher, suggesting a weaker relationship. Post hoc analysis for these parameters showed differentiation of the pathology negative regions from those with cancer findings (Figure 7). Only F_p_ and E showed clear monotonic progression of values in function of increasing GG, but generally the GG = 4 lesions had parameter values more clearly distinct from the pathology negative regions than the GG = 1 lesions (Figure 7, Appendix A). Due to the small number of biopsy samples (N = 6), the association with GG for the TZ was not examined.

## 4. Discussion

In this pilot study, we adopted an ultrafast prostate DCE-MRI acquisition (1.695 s/dynamic) as a basis for examining the early (first 2 min) enhancement patterns in healthy prostate tissues and prostate lesions and their relationships with prostate zone, PI-RADS category, and biopsy GG.

We found differences between radiologically normal-appearing PZ and TZ only for the 2CU model parameter F_p_ and for the exponential EMM parameter α (Table 5). In the two-way ANOVA of all ROIs, considering prostate zone and PI-RADS category as factors, however, F_p_ was no longer seen to differentiate between zones, but α and the 2CU model parameter E, along with the sigmoidal EMM parameters A_1_ − T_0_ and A_2_, did so (Table 6). The 2CU model parameters F_p_, MTT_p_, T_c_, and E, exponential EMM parameter α, sigmoid EMM parameters A_1_ − T_0_ and A_2_, and the curve shape parameter TTP all showed associations with PI-RADS categories (Table 6). These same parameters showed associations with GG in the PZ, but only when the radiologically normal-appearing ROIs were included as part of the non-cancer GG (Table 7). When limited only to the biopsy samples, none of the parameters showed significant associations with G.

As PI-RADS category evaluations have generally been shown to correlate with Gleason score and GG [19,20], the fact that the parameters associated with PI-RADS categories also showed associations with GG (in the PZ) is reassuring. In addition, post hoc testing for those parameters showing differences in the above ANOVAs showed a tendency for the model parameters to progress monotonically (either increasing or decreasing) with increasing PI-RADS category or GG (Figure 6 and Figure 7). Thus, differences relative to normal-appearing tissue tended to be larger for higher PI-RADS categories 4 and 5 than for PI-RADS category 3 lesions. This tendency, however, was less pronounced for the transitional/central zone lesions than peripheral zone lesions, possibly because of the smaller number of lesions involved. 

The early enhancement phase examined in the present study captures the key feature used in radiological assessment of prostate DCE-MRI under PI-RADS v2.1, namely the presence (or not) of early enhancement in suspected peripheral zone lesions. The closer association of the measured parameters with PI-RADS categories than with GG may reflect tissue alterations, such as changing voxel occupancy by cells that also impact the relaxation time and diffusion properties used in the radiological PI-RADS assessment [1]. Assessment of GG, on the other hand, depends on the visual appearance of the cells rather than their density [18], which may be less closely correlated with 2CU model parameters. A concern with the comparison with GG is that significant associations were only seen when samples from radiologically normal-appearing tissues were included, for which pathology assessment was not available but assumed to be GG 1. The correlations may therefore reflect differences between normal and abnormal prostate of potential use in diagnosis rather than between GGs, which would be of interest for lesion staging.

Notably, PS and K_trans_ failed to show significant associations with prostate zone, PI-RADS category, or GG. Some previous studies have suggested that, on average, K_trans_ values are higher in cancer than in healthy prostate tissue [21,22,23] but with wide ranges of values for both. The lack of effect for PS and K_trans_ may be due to our limitation of the data window to the first 2 min of enhancement not being appropriate for the PS fitting in the 2CU model or our choice not to limit v_p_ to plausible physiological values as we did not rescale via arterial input function. Either of these may have introduced a systematic error in the fitting. We note that a recent systematic review indicated K_trans_ as the Tofts model parameter having the highest correlation with Gleason score but that the level of correlation was too small for clinical use [24].

Generally, erroneous assumptions about the compartmental structure of the prostatic tissues may be expected to lead to systematic errors in physiologic parameters [2,25]. The 2CU model is based on the assumption of gadolinium redistribution in two compartments (intravascular and extravascular extracellular compartments). While the 2CU model yielded several parameters of interest for the prediction of prostate cancer from DCE-MRI, this pilot study does not provide insight into whether the 2CU model is widely applicable to the prostate. Some authors have proposed a differential uptake of contrast by different prostate tissue components [26,27], considering that a lack of basal cells could render the cancerous glandular lumen more permeable to gadolinium-based contrast agents than in healthy glands. This assumption is critical because, if the contrast agent is also redistributed in the glandular lumen, it would represent a third compartment, and more complex compartmental patterns would be required for DCE-MRI analysis of prostate tissue [26]. We analyzed data from the first two minutes of enhancement to mitigate the influence of wash-out and differential contrast uptake by different prostate tissue components on fitting.

The exponential EMM parameter α, sigmoidal EMM parameters A_1_ − T_0_ and A_2_, and the curve shape descriptor TTP that had significant associations with both PI-RADS category and GG all provide indications of the time of onset or the time between onset and maximum of a rising curve. Again, these are understandably related to the early and maximum enhancement characteristics in PI-RADS evaluation of DCE-MRI, with the EMM and curve parameters being more closely associated with PI-RADS status than the 2CU model parameters (lower *p*-values). This may support the use of either very simple descriptors (the exponential EMM or curve shape TTP) or the 2CU model as candidates for improving the contribution of DCE-MRI to prostate cancer diagnosis, though the association with GG were weaker.

The limited contribution that DCE-MRI makes to PCa diagnosis in the majority of cases under PI-RADS guidelines has led many to question whether the additional costs (contrast agent, preparation, scan, and reporting time) are warranted [28,29,30,31]. Unfortunately, this small pilot study is not sufficient to provide indications for use in problematic areas such as differentiating benign hyperplasia from carcinoma in the transitional zone or prostatitis from carcinoma in the peripheral zone [32]. Our results do, however, suggest that several parameters describing early contrast uptake, including F_p_, MTT_p_, T_c_, E, α, A_1_ − T_0_, A_2_, and TTP calculated on the basis of high temporal resolution DCE-MRI, are all likely to be more useful in these situations than conventional measures of perfusion such as K_trans_ and PS. One possible path forward would be to incorporate in-line analysis of non-contrast sequences (T2- and DWI-weighted) and to perform high-temporal resolution DCE-MRI only in those patients with intermediate levels of certainty (PI-RADS categories 2–4) [33]. Further studies of the correlation of the derived enhancement parameter with biopsy findings would then be concentrated in the population of patients of greatest clinical relevance. 

We recognize a number of procedural limitations in our study. Manual delineation of regions of interest (ROIs) could introduce bias due to operator-dependent delineation errors. Moreover, the ROIs for normal-appearing prostate specifically excluded PI-RADS category 3 or higher lesions but included regions that would be considered PI-RADS category 1 or 2 and thus suspected of being subject to pathology but not cancer. Prostate zone and lesion delineation is a rapidly developing research area, with a growing number of machine learning algorithms being presented [34,35,36,37], but the adaptation and retraining required for specific scanners and protocols were considered unwarranted in this small cohort. In routine use for a large population, such automated solutions would be a valid alternative. Further, we did not perform baseline T1-mapping of prostate tissues in the MRI protocol, which could introduce bias in the calculation of semi-quantitative or quantitative parameters, as malignant lesions typically have lower T1 values than healthy tissues. A well-validated widely available method for accurate prostate T1-mapping is still lacking. 

There are also several aspects regarding the mathematical models that warrant further study. First, we concentrated on the early phase of contrast enhancement because this most closely relates to the characteristics used radiologically in the current subjective evaluation of DCE-MRI under PI-RADS. This choice may limit the quality of fitting of some models (e.g., 2CXM, Tofts models), but has been demonstrated to allow acceptable performance of the 2CU or Patlak models [25]. Second, there are other possibilities for both quantitative (pharmacokinetic) and semi-quantitative models (EMMs) and dispersion imaging [38] that may be well suited for describing the specific contrast dynamics occurring in the early enhancement phase of prostate DCE-MRI. Lastly, we did not employ a model selection criterion (e.g., minimum sum of squares error or Akaike information criteria [8,39]) to determine the best-fitting model. This was considered excessive in this pilot study but will be justified in a larger study involving a wider range of models.

## 5. Conclusions

Our findings in this pilot study using high temporal resolution prostate DCE-MRI suggest that characterization of early enhancement, whether by the 2CU model, exponential or sigmoid EMMs, or even the simple curve shape TTP descriptor, can provide parameters that relate to PI-RADS category and GG. This reinforces recent demonstrations that DCE-MRI of the prostate with a temporal resolution of less than 2 s is feasible. Further study is needed to show whether these strategies improve diagnostic performance relative to the existing PI-RADS criteria. At a time when there are calls to reduce contrast agent use in general and to further limit its role in prostate cancer diagnosis in particular, such studies would provide needed insight into the most effective use of DCE-MRI.

## Figures and Tables

**Figure 1 diagnostics-14-00870-f001:**
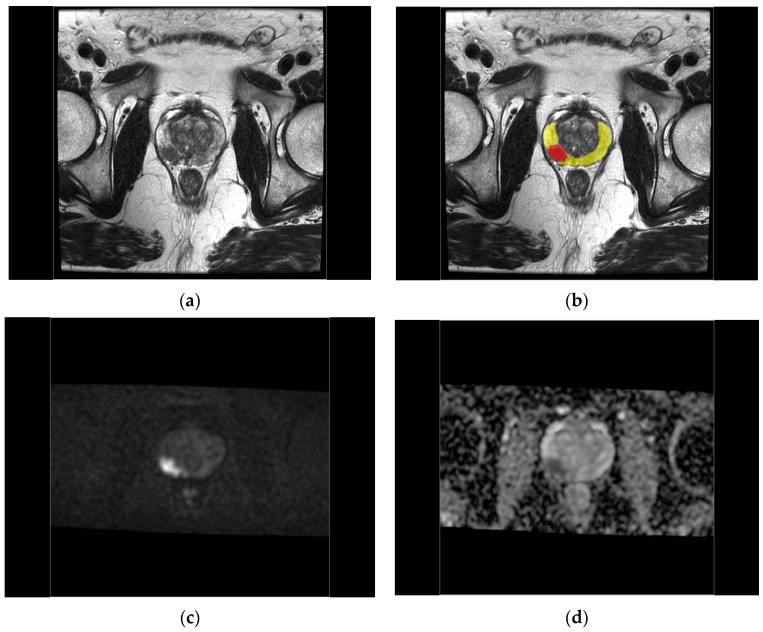
Male 59 years with PIRADS category 4, Gleason score 4 + 4 peripheral zone lesion. (**a**) The anatomical T2 image with (**b**) ROIs defined manually for the zone lesion (red ROI) and the adjacent healthy-appearing peripheral zone (yellow ROI). The lesion is also seen as a hyperintensity in (**c**) the co-registered high b-value DWI, with (**d**) low ADC, (**e**) early enhancement in native, and (**f**) subtraction ultrafast DCE-MRI images.

**Figure 2 diagnostics-14-00870-f002:**
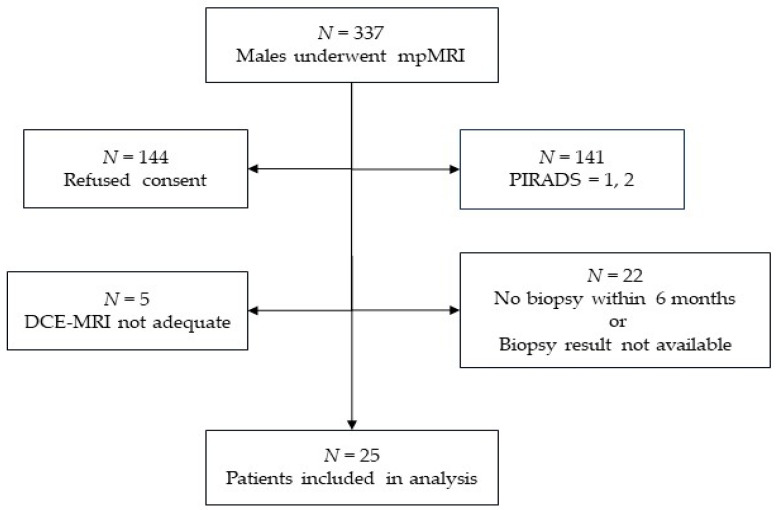
Flowchart breakdown of the bases for patient exclusion and the total included in the study.

**Figure 3 diagnostics-14-00870-f003:**
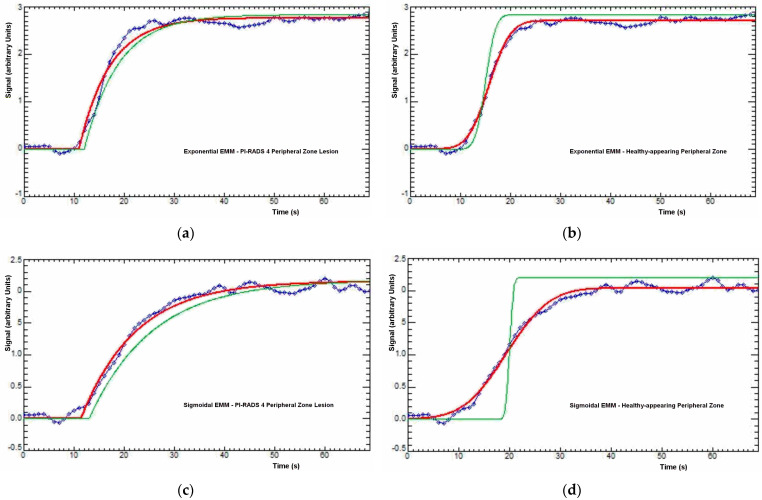
Example curve fittings for the exponential (top (**a**,**b**)) and sigmoidal (bottom (**c**,**d**)) empirical mathematical models in a peripheral zone PI-RADS category 4 lesion voxel (left (**a**,**c**)) and healthy-appearing peripheral zone voxel (right (**b**,**d**)) from a 55-year-old prostate cancer patient. Biopsy of the lesion revealed a Gleason score 3 + 3 lesion. Green = starting estimate curve used for search initialization. Red = final fitted model.

**Figure 4 diagnostics-14-00870-f004:**
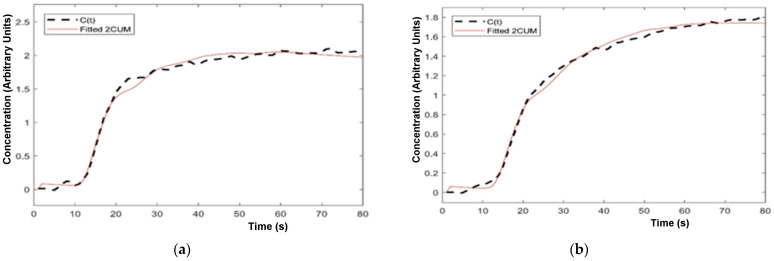
Example curve fitting for the 2CUM in a peripheral zone PI-RADS 4 lesion ROI (left: (**a**)) and healthy-appearing peripheral zone ROI (right: (**b**)), from a 59-year-old PCa patient. Biopsy of the lesion revealed a Gleason 4+4 lesion.

**Figure 5 diagnostics-14-00870-f005:**
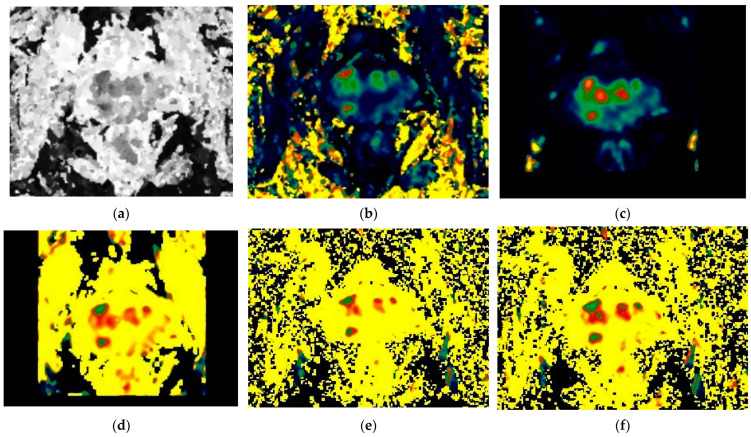
Maps of (**a**) TTP, (**b**) α, (**c**) F_p_, (**d**) MTT_p_, (**e**) A_2_, and (**f**) A_1_ − T_0_, along with conventional PI-RADS images: (**g**) T2-weighted, (**h**) synthetic high b-value DWI (2000 s/mm^2^), and (**i**) ADC map, for a patient with peripheral zone and transition zone lesions. The T2- and DWI-weighted images allow a PI-RADS 4 categorization of the Gleason score 3 + 4 peripheral zone lesion (thin arrow) in light of only limited reduction in ADC, but only a PI-RADS 3 categorization of the Gleason score 3 + 4 transition zone lesion (thick arrow). A region of benign prostatic hyperplasia in the transition zone at MRI-guided prostate biopsy (arrowhead) had less extreme values in the parameter maps than the lesions. Increasing values in the colorized maps are represented by the color scale progression blue, green, red, and yellow.

**Figure 6 diagnostics-14-00870-f006:**
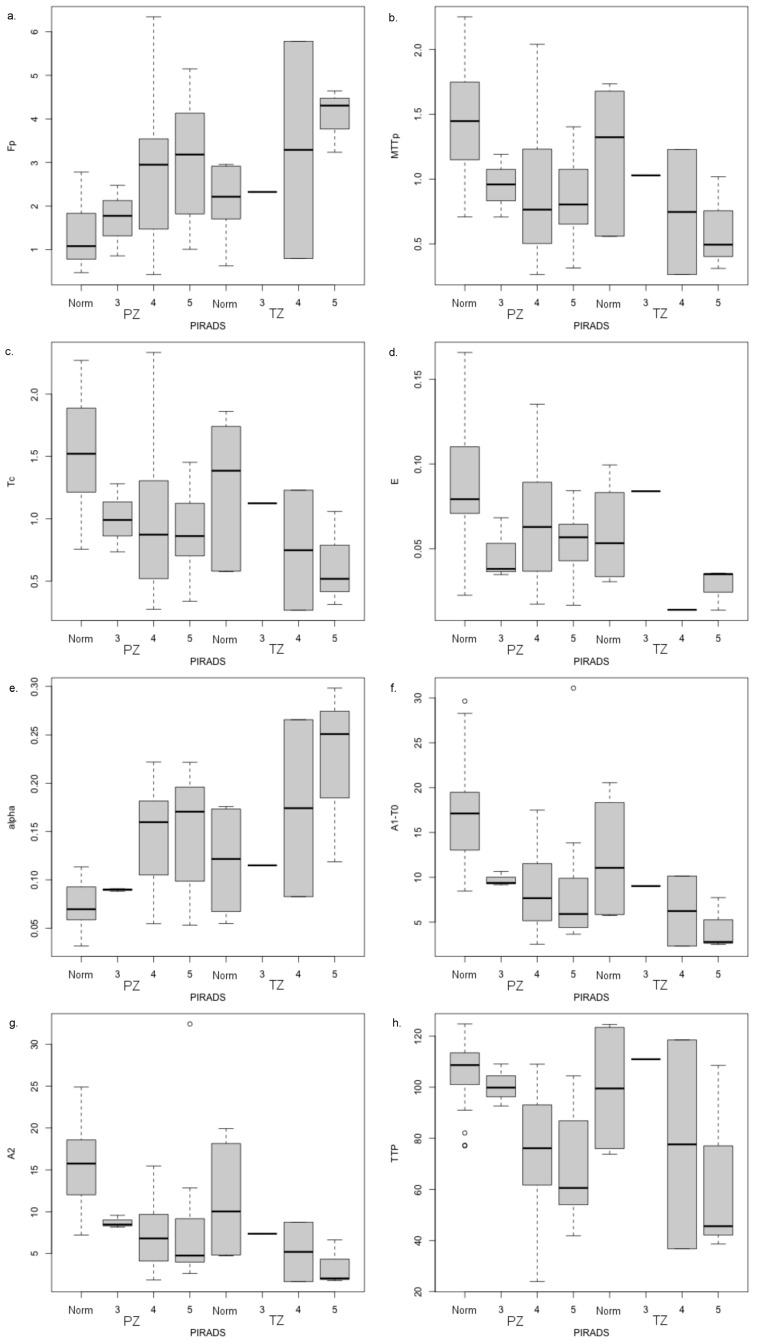
Post hoc comparison by tissue type and PI-RADS category for (**a**) F_p_, (**b**) MTT_p_, (**c**) T_c_, (**d**) E, (**e**) α, (**f**) A_1_ − T_0_, (**g**) A_2_, and (**h**) TTP, which showed significant differences in two-way ANOVA between normal-appearing (Norm) tissues and PI-RADS category 3, 4, and 5 lesions.

**Figure 7 diagnostics-14-00870-f007:**
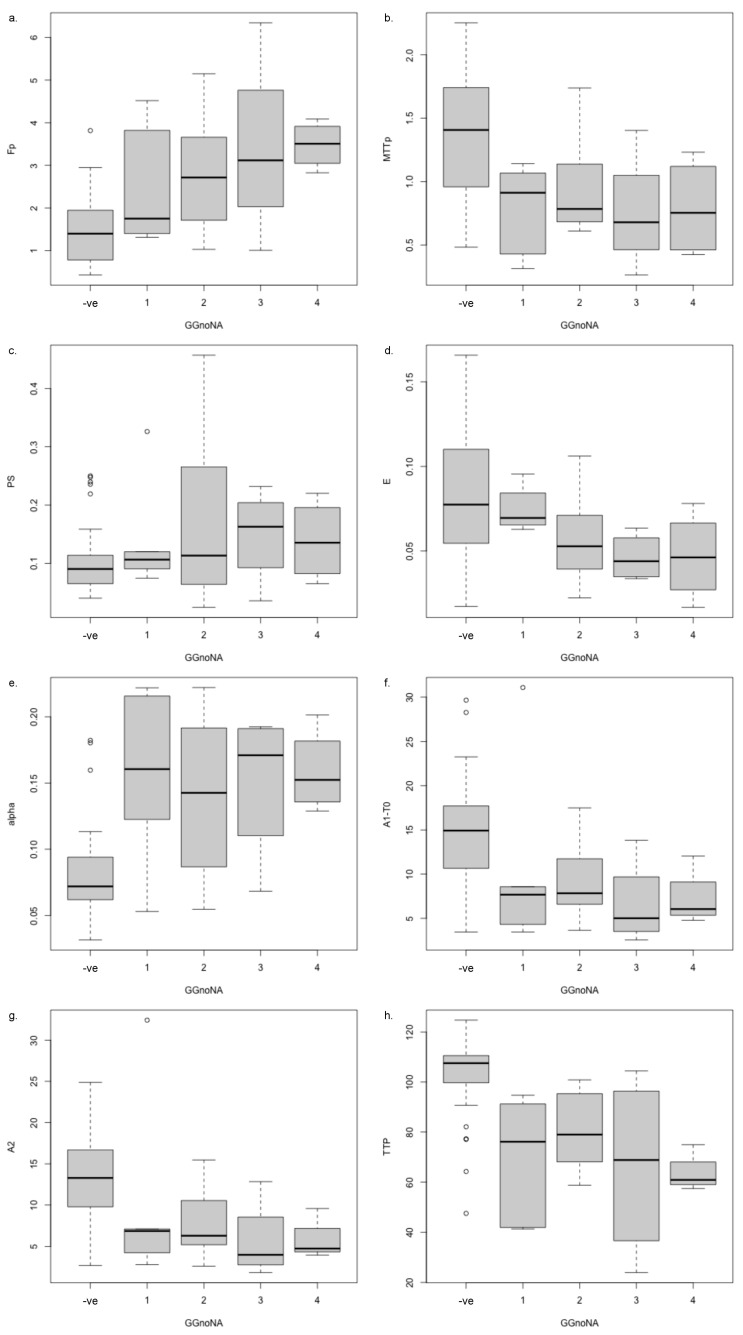
Post hoc analysis of differences seen for (**a**–**d**) 2CU model parameters F_p_, MTT_p_, T_c_, and E; (**e**) exponential EMM parameter α; (**f**,**g**) sigmoidal EMM parameters A_1_ − T_0_ and A_2_; and (**h**) curve shape TTP, seen to be different in one-way ANOVA with GG (normal-appearing or biopsy negative vs. GG1, GG2, GG3, and GG4).

**Table 1 diagnostics-14-00870-t001:** Comparison of pharmacokinetic model parameters *: Tofts (T), extended Tofts (ET), Patlak, and two-compartment uptake (2CU) models.

Pharmacokinetic Parameter	Model			
	T	ET	Patlak	2CU
Plasma volume fraction		**v_p_**	**v_p_**	**v_p_**
Extravascular extracellular volume fraction	**v_e_**	**v_e_**		
Plasma flow				**F_p_**
Transfer constant	**K_trans_**	**K_trans_**	**K_trans_**	K_trans_ = EF_p_
Rate constant	k_ep_ = K_trans_/v_e_	k_ep_ = K_trans_/v_e_		
Permeability surface area product				**PS**
Extraction fraction				E = PS/(PS + F_p_)
Plasma mean transit time				MTT_p_ = v_p_/(PS + F_p_)
Capillary transit time				T_c_ = v_p_/F_p_

* Bold = parameters involved in fitting.

**Table 2 diagnostics-14-00870-t002:** Ultrafast dynamic contrast enhanced prostate MRI sequence parameters.

Parameter	Value
TE	0.96 ms
TR	2.871 ms
Flip Angle	10°
Field of View	180 mm
Acquisition Matrix	110 × 110
Reconstruction Matrix	256 × 256
In-Plane Resolution	0.703 × 0.703 mm
Bandwidth per Pixel	41.67 Hz
Acceleration Factor	1.75 × 2
Slice Thickness	3 mm
Number of Slices	26
Time per Dynamic	1.695 s
Number of Dynamics	150
Duration	4:14 min:s

**Table 3 diagnostics-14-00870-t003:** Calculated parameters and their units of measure and range of values accepted in fitting.

Model	Parameter	Unit of Measure	Range
2CU	v_p_	none (mL/mL)	[0,10]
	PS	min^−1^ (mL/min/mL)	(0,10]
	F_p_	min^−1^ (mL/min/mL)	[0,10]
	T_c_	min	[0,5)
	MTT_p_	min	[0,5)
	E	none	[0,1]
	K_trans_	min^−1^ (mL/min/mL)	[0,10]
Exponential EMM	A	none	none
	α	time^−1^ (# of dynamics)^−1^	(0–1]
	t_0_	time (# of dynamics)	none
Sigmoidal EMM	A_0_	none	none
	A_1_ − T_0_	time (# of dynamics)	(0,100]
	A_2_	time (# of dynamics)	(0,100]
Curve Shape	TTP	time (# of dynamics)	None

Shaded parameters were not included in analyses. Parentheses [ and ] indicate inclusive of the lower and upper value, respectively; parentheses ( and ) indicate exclusive of the lower and upper value, respectively.

**Table 4 diagnostics-14-00870-t004:** Summary of patient, ROI, and biopsy characteristics.

	Characteristic	Value Mean (Range) or Count (%)
Patients (*N* = 25)	Age (years)	67 (55–84)
	PI-RADS ≥ 3 lesions per patient	
	1	16 (64%)
	2	8 (32%)
	3	1 (4%)
Region of interest:	PI-RADS Category	
Lesion (*N* = 35)	3	4 (11.4%)
	4	21 (60.0%)
	5	10 (28.6%)
Healthy Tissue (*N* = 35)	assumed ≤ 1	35 (100%)
	Prostate Zone	
	Peripheral Zone	29 (82.9%)
	Transition Zone	6 (17.1%)
Biopsy Samples (*N* = 35)	Grade Group	
	negative	11 (31.4%)
	1	6 (17.2%)
	2	9 (25.7%)
	3	5 (14.3%)
	4	4 (11.4%)

**Table 5 diagnostics-14-00870-t005:** Comparison of model parameters in radiologically normal-appearing prostate zones ^+^.

Model	Parameter	Prostate Zone		
		PZ	TZ	*t*-Test *p*-Values ^
		Mean Value		
2CU	v_p_	1.761	1.912	0.684
	PS	0.108	0.094	0.597
	F_p_	1.363	2.105	0.0245
	MTT_p_	1.439	1.196	0.235
	T_c_	1.534	1.254	0.195
	E	0.0861	0.0589	0.0512
	K_trans_	0.0938	0.0839	0.657
Exponential EMM	α	0.0723	0.119	2.10 × 10^−3^
Sigmoidal EMM	A_1_ − T_0_	16.671	12.099	0.0724
	A_2_	15.071	11.287	0.0932

^+^ Shaded values indicate significance at a *p* < 0.05 level after Benjamini–Hochberg correction. ^^^ *t*-test between peripheral zone (PZ) and transition/central zone (TZ).

**Table 6 diagnostics-14-00870-t006:** Two-way ANOVA of models with prostate zone and PI-RADS category as factors ^+^.

Model	Parameter	Radiological Evaluation	
		ANOVA *p*-Values by Factor °	
		Prostate Zone	PI-RADS Category
2CU	v_p_	0.866	0.418
	PS	0.348	0.443
	F_p_	0.403	2.72 × 10^−5^
	MTT_p_	0.151	1.45 × 10^−4^
	T_c_	0.118	1.53 × 10^−4^
	E	0.0236	6.99 × 10^−3^
	K_trans_	0.374	0.298
Exponential EMM	α	2.43 × 10^−3^	2.44 × 10^−7^
Sigmoidal EMM	A_1_ − T_0_	0.0264	4.28 × 10^−6^
	A_2_	0.0321	4.94 × 10^−6^
Curve Shape	TTP	0.605	4.88 × 10^−7^

^+^ Shaded values indicate significance at a *p* < 0.05 level after Benjamini–Hochberg correction for multiple comparisons. ° Prostate zone: 2 levels (peripheral zone (PZ) and transition/central zone (TZ)); PI-RADS category: 4 levels (normal = radiologist chosen as normal-appearing and PI-RADS categories 3, 4, and 5).

**Table 7 diagnostics-14-00870-t007:** Comparisons via *t*-test and one-way ANOVA of 2CU model, EMM, and time to peak parameters according to pathology grade group ^+^.

Model	Parameter	Peripheral Zone	
		Biopsy Only	Biopsy Sample + Normal-Appearing
		*t*-Test *p*-Value ^	One-Way ANOVA *p*-Value °
		−ve vs. +ve	−ve, GG1, GG2, GG3, GG4
2CU	v_p_	0.355	0.236
	PS	0.186	0.318
	F_p_	0.0527	9.41 × 10^−5^
	MTT_p_	0.117	0.002
	T_c_	0.0995	1.73 × 10^−3^
	E	0.391	0.0346
	K_trans_	0.168	0.234
Exponential EMM	α	0.153	4.1 × 10^−5^
Sigmoidal EMM	A_1_ − T_0_	0.950	8.73 × 10^−3^
	A_2_	0.940	7.03 × 10^−3^
Curve Shape	TTP	0.0608	4.72 × 10^−6^

^+^ Shaded values indicate significance at a *p* < 0.05 level after Benjamini–Hochberg correction for multiple comparisons. ^ *t*-test between positive (+ve; GG ≥ 1) and negative pathology findings of cancer in peripheral zone biopsy samples. ° GG: 5 levels: normal (normal-appearing prostate ROI or negative pathology finding) and GG values (1, 2, 3, and 4).

## Data Availability

The data presented in this study are available on request from the corresponding author.

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
