# Peer review of "Comparison of Early Contrast Enhancement Models in Ultrafast Dynamic Contrast-Enhanced Magnetic Resonance Imaging of Prostate Cancer"

_diagnostics, 2024, doi:10.3390/diagnostics14090870_

Round 1
Reviewer 1 Report
Comments and Suggestions for Authors
The authors report an ultrafast DCE-MRI acquisition for investigating early enhancement profiles in healthy adult tissues and prostate lesions. Various parameters are used to study the relationship between the lesion status and prostate zone, PI-RADS category, and grade group. The preliminary results show a less than 2-second temporal resolution is feasible to study prostate cancer with the first two-minute enhancement parameters. The authors need to address the following issues before the manuscript is accepted for publication in Diagnostics:
- In Figure 3, please clarify the rationale for “initial guess” for search initialization.
- Please clarify the patient consent information for the retrospective study and confirm that the patients consented to the use of their data for research during enrollment.
- The authors need to polish the manuscript in errors of grammar, such as “Un-surprisingly,” “That the parameters showing…” and “it having previously been assumed that…”
- Please pay attention to the format and consistency, such as DCE MRI vs. DCE-MRI, arterial input function (AIF) vs. arterial input function (AIF(t)), Table 3 legend, and references 30 – 33.
The manuscript needs extensive grammar polish.
Reviewer 2 Report
Comments and Suggestions for Authors
This study is reinforces that DCE MRI could be useful (in selected patients) and improve diagnostic performance witin the existing framework of the PI-RADS criteria.
1. Reduction of intravenous administration of MR-contrast agents is an issue indeed, as is the preparation time (nursing cost), cost for the contrast agent, cost for the disposables to be used, the extended measuring time on the MR-equipment, the cost for the soft ware package, and the time interpretation/reading.
2. It is not fully clear from the study for which patiens (prostatic lesions) fast DCEI is beneficial.
Would the described technique be able to differentiate benign hyperplastic nodules in the transitional zone form (well) differentiated prostatic transitional zone carcinoma ?
Would the describe technique be able to differentiate peripheral zone (granulomatous) prostatitis from carcinoma ?
The illustration (figure) used is not truly convincing for the technology described: indeed, this is a typical nodular T2-hypointense en diffusion restrictive lesion, a typical peripheral zone carcinoma.
More convincingly would be to illustrate the technique with a more challenging case, as is encountered regularly in routine practice: as mentioned above, a case with pronounced peripheral zone prostatitis/fibrosis + carcinoma and/or a case with multinodular hyperplasia of the transitional zone + transitional zone carcinoma.
In the discussion section, reflection on how to implement the described technique of DCEI: in all patients referred to MRI? If not, how to selct te patients? Would it be advisable to use the technique for future MRI-based screening programs for prostate cancer?
Round 2
Reviewer 2 Report
Comments and Suggestions for Authors
No additional comments.